# The Elephant in the Room: A Systematic Review of the Application and Effects of Psychological Treatments for Pregnant Women with Dual Pathology (Mental Health and Substance-Related Disorders)

**DOI:** 10.3390/ijerph21040392

**Published:** 2024-03-23

**Authors:** Irene Caro-Cañizares, Nayara López Carpintero, Rodrigo Carmona-Camacho

**Affiliations:** 1Facultad de Ciencias de la Salud y la Educación, Universidad a Distancia de Madrid, UDIMA, 28400 Collado Villalba, Spain; 2Departamento de Obstetricia y Ginecología, Hospital Universitario del Tajo, 28300 Aranjuez, Spain; 3Departamento de Psiquiatría, Fundación Jiménez Díaz, 28040 Madrid, Spain

**Keywords:** pregnancy, mental health, substance use, dual pathology, systematic review

## Abstract

Purpose: Maternal mental health and substance use, referred to as dual pathology, represent significant concerns associated with adverse pregnancy and birth outcomes, a prevalence higher than commonly anticipated. Nonetheless, a notable dearth exists ofevidence-based treatment protocols tailored for pregnant women with dual pathology. Methods: A systematic review, adhering to the PRISMA methodology, was conducted. Results: Out of the 57 identified papers deemed potentially relevant, only 2were ultimately included. Given the limited number of studies assessing the efficacy of psychological interventions utilizing randomized controlled trials (RCTs) for both mental health and substance misuse, and considering the diverse objectives and measures employed, definitive conclusions regarding the effectiveness of psychological interventions in this domain prove challenging. Conclusions: Maternal mental health appears to be the proverbial “elephant in the room”. The development of specialized and integrated interventions stands as an imperative to effectively address this pressing issue. As elucidated in the present review, these interventions ought to be grounded in empirical evidence. Furthermore, it is essential that such interventions undergo rigorous evaluation through RCTs to ascertain their efficacy levels. Ultimately, the provision of these interventions by psychology/psychiatric professionals, both within clinical practice and the RCTs themselves, is recommended to facilitate the generalizability of the results to specialized settings.

## 1. Introduction

The perinatal period, spanning from pregnancy through to the postpartum period, represents a pivotal time of profound physiological, psychological, and social changes for both the mother and the developing baby [1]. This period is characterized by dynamic processes of fetal growth and development, maternal physiological adaptations, and significant emotional transitions as individuals prepare for parenthood [2]. It is during this critical window that the foundation for lifelong health and well-being is established, making the mental health and well-being of pregnant women paramount.

Indeed, the mental health and emotional well-being of expectant mothers play a crucial role in shaping the trajectory of pregnancy outcomes. Mental health issues, including depression, anxiety, and other disorders, are recognized as leading causes of disability among pregnant women, exerting profound effects on maternal health and pregnancy outcomes [3]. Left untreated, these mental health challenges can escalate andpose significant risks, including increased maternal mortality rates, adverse birth outcomes, and long-term implications for the physical and emotional well-being of both the mother and the baby [3]. Perinatal exposure to maternal mental health problems has been associated with adverse fetal outcomes, including preterm births; adverse effects on cognitive, behavioral, and psychomotor development; and mental health disorders in children [4,5,6], highlighting the interdependence of maternal and child health during the perinatal period. Additionally, maternal mental disorders during the perinatal period can lead to increased healthcare system utilization and associated social costs [7].

The simultaneous occurrence of mental health disorders and substance use problems, commonly referred to as dual pathology, represents a significant complicating factor in the management of maternal health during pregnancy. This co-occurrence not only exacerbates the severity of individual mental health and substance use issues but also introduces unique challenges and complexities in clinical intervention and treatment [8]. Dual pathology during pregnancy poses a heightened risk for adverse maternal and fetal outcomes, compounding the already considerable health risks associated with each condition independently [9]. The presence of co-occurring mental health disorders and substance use problems is often associated with a more severe clinical presentation, including increased symptom severity, greater functional impairment, and higher rates of comorbidity with other medical conditions [8].

In addition, smoking, benzodiazepine use, or other forms of substance use during pregnancy have bidirectional and deleterious interactions with mental health problems. For example, substance use may exacerbate the symptoms of depression or anxiety, while untreated mental health issues may contribute to ongoing substance use and relapse. It is also noteworthy that substance use represents the primary preventable cause of mortality and morbidity during pregnancy [10]. The magnitude of substance use problems during pregnancy is notable, as evidenced, among other examples, by smoking prevalence ranging from 10 to 19% in high-income countries [9,11]. In a relatively recent systematic review, the average prevalence of drug use calculated in pregnant women was 1.65% among the studies that conducted questionnaires and 12.28% in studies that performed toxicological analysis [12]. Regarding alcohol, the estimated percentage of prenatal use is approximately between 8 and15% [13,14], while cannabis use percentage is estimated to be between 1 and 2% [15]. This substance use is even more problematic considering that women who use substances during pregnancy typically use more than one, multiplying the risk to the fetus [16]. Prenatal exposure to cocaine, alcohol, or tobacco is known to have numerous negative effects on children, including adverse pregnancy and birth outcomes such as low birth weight, miscarriage, prematurity, congenital abnormalities, and neonatal or sudden infant death [17,18,19,20,21].

Despite the challenging scenario outlined above, pregnancy and maternity are periods generally associated with higher levels of motivation to seek clinical attention in a broader sense [22]. These pivotal life stages serve as powerful catalysts for change and can significantly influence women’s attitudes and behaviors towards seeking healthcare services, addressing underlying health issues, and acting as strong motivators for drug abuse cessation [23]. Therefore, it is imperative for healthcare providers to capitalize on this intrinsic motivation and readiness for change by offering timely and tailored therapeutic support to pregnant women seeking assistance. Consequently, as Howard and Khalifeh [3] stated, the World Health Organization (WHO) has emphasized the urgent need for “evidence-based, cost-effective, and human rights-oriented mental health and social care services in community-based settings for the early identification and management of maternal mental disorders”. By providing comprehensive and evidence-based interventions that address the complex interplay of mental health and substance use concerns, healthcare providers can help these pregnant women, optimize maternal and fetal outcomes, and foster healthier pregnancies and transitions to motherhood.

In the literature [24], three different approaches for treating dual pathology are outlined: (1) sequential treatment, in which one pathology is addressed, first followed by the other, which has limited support due to the challenges of prioritizing which problem to address first and difficulties in follow-up; (2) parallel treatment, where each pathology is addressed by different health resources and professionals simultaneously, which is the most common model but presents challenges in coordinating resources and building therapeutic adherence; and (3) integrated treatment, where both pathologies are addressed concurrently and using a multidisciplinary approach by the same team of professionals and resource, which is the most recommended approach but is less widespread due to the complexities of offering integrated resources within healthcare systems.

This integrated approach makes it possible to address the complexity of dual pathology cases, improving the remission of symptoms, as well as substance use cessation. The literature points out the need for a multidisciplinary and comprehensive healthcare approach to these patients due to the complexity of their medical, psychological, and social problems [8,25,26], emphasizing the need to involve and train different professionals and coordinate interventions across institutions to improve care for substance-using women and their children [27].

However, there is a lack of evidence-based treatment algorithms for pregnant women with dual pathology [28]. While there are some recent systematic reviews of mental health interventions for pregnant women (e.g., [3,28,29]), the information on substance use treatment is less extensive and often focuses solely on alcohol [30] or tobacco use [31] or specific delivery modalities [32]. Furthermore, information on dual pathology is practically nonexistent.

Given the existing gap in the research, it is imperative to conduct systematic explorations of therapeutic interventions aimed at evaluating their efficacy in addressing dual pathology among pregnant women. This investigation will contribute to filling the knowledge void and inform evidence-based approaches to improving the mental health outcomes of pregnant women facing dual pathology challenges. As part of the WOMAP (Woman Mental Health and Addictions on Pregnancy) initiative, a review was conducted following the PRISMA [33] methodology for systematic reviews.

## 2. Method

The systematic review protocol was registered in PROSPERO in May 2022 [CRD42022330542]. The research question guiding the review was as follows: What are the published applications and the effects of psychological interventions assessed in clinical trials for pregnant women with dual pathology (i.e., the coexistence of a mental health and a substance use disorder)?

The sources were PubMed (MEDLINE), through Sysrev.com, and ProQuest. The restrictions included language (English and Spanish) and source type (scientific journal).

The inclusion criteria were:Type of study: clinical trials, quasi-experimental and experimental studies.Intervention condition: any type of psychological intervention, psychological treatment, psychotherapeutic technique, or psychoeducational intervention.Participants: pregnant women (aged >17)with dual pathology (the coexistence of a mental health and a substance use disorder).

The exclusion criteria were:Publications of clinical trial protocols that didnot include efficacy/effectiveness data.Research that did not address the coexistence of a mental health disorder and a substance use disorder in an integrated way.An intervention condition different from a psychological intervention, psychological treatment, psychotherapeutic technique, or psychoeducational intervention (e.g., pharmacological treatment, acupuncture, spiritual support, etc.).Participants: Adolescents (under 18 years old) and women who were not pregnant at the time of the clinical trial. Studies were not excluded if they had at least some participants who were 18 or older and pregnant.

The search string was:

ti((pregnancy OR “pregnant woman” OR “pregnant women” OR “prenatal care”)) AND (“clinical trial” OR “evidence based intervention” OR “effectiveness of intervention” OR “evaluation of effectiveness” OR “randomized clinical trial” OR “randomized controlled trial” OR “RCT” OR “recruitment of patients” OR “trial design” OR “quasi-experimental study” OR “experimental study”) AND (“treatment” OR “intervention” OR “psychotherapy” OR “psychoeducation” OR “cognitive behavioral therapy” OR “cognitive behavioral intervention” OR “CBT”) AND (“mental health” OR “depression” OR “major depression” OR “neurotic depression” OR “unipolar depression” OR “reactive depression” OR “dysthymia” OR “chronic depression” OR “emotional depression” OR “anxiety” OR “anxiety disorders” OR “anxiety neurosis” OR “posttraumatic stress” OR “posttraumatic stress disorder” OR “emotional trauma” OR “acute stress disorder” OR “trauma” OR “PTSD” OR “stress reactions” OR “adjustment disorders” OR “stress and trauma related disorders”) AND (“substance abuse” OR “substance related disorders” OR “addictive disorders” OR “substance use disorder” OR “substance use” OR “substance addiction”).

### Data Extraction

The search was conducted until 11 May 2022 and provided 46 results from PubMed and 399 results from ProQuest. After removing duplicates, 386 results remained.

The selection of relevant studies followed a three-phase process conducted by three reviewers (see Figure 1): first, the titles and abstracts of all the identified papers were reviewed based on the previously established inclusion and exclusion criteria. In the second phase, the full texts of the works identified as potentially relevant (50 papers) were reviewed. This phase was completed with a cross-reference check until 27 April 2023, resulting in the extraction of another 6 studies and their inclusion for full-text review. Once the inclusion and exclusion criteria had been applied, only two papers were finally selected. In the third phase, relevant information was extracted. Finally, prior to the publication of this manuscript, in February 2024, the search was conducted again to lookfor newly published papers, and after reviewing the title and abstract of 110 new references, no new results were found. A cross-reference check until March 2024 resulted in one extra study included for full-text review (see Table 1).

The process of searching for, analyzing, and selecting the studies was carried out by two reviewers (I.C.C. and N.L.C.), while the third reviewer (R.C.C.) checked the decisions. Disagreements were resolved through discussion and consensus.

The information collected included each study’s type, design, methodology, sample size, sample characteristics, timing and number of evaluations, and effect sizes.

The entire decision-making process was recorded in Sysrev.com, accessed on 20 February 2024.

## 3. Results

In the two papers ultimately included in this review (see Table 2), the total sample consisted of 1275 pregnant women, with individual study sample sizes ranging from 322 participants in Barlow et al.’s study [39] to 953 participants in Joseph et al.’s study [51]. Both studies were conducted in the USA among women who were members of racial/ethnic minority groups (African American, Latina, and Native American) recruited from different Health Service clinics. The mean ages of the participating women referred to in the papers ranged from 18.1 [39] to 24.8 years old [51]. In Barlow et al.’s study [39], 186 women (57.8%) were at least 18 years old, while all the participants in Joseph et al.’s study [51] were 18 years old or above. Regarding the methodology, both studies were randomized trials with participants randomly assigned to one of two groups: an experimental group versus either a treatment-as-usual (TAU) group [51] or an optimized standard care group [39].

The mental health problems evaluated include depressive symptoms and internalizing-related problems (anxiety). The substance use problems include alcohol abuse, tobacco smoking, and the use of marijuana and other illegal drugs.

The psychological interventions assessed could be described as psychoeducational interventions [39] or interventions based on some type of counseling or individual support [51], provided by “pregnancy advisors” or “family health educators”. In both cases, these interventions were adapted to specific social groups (Native American pregnant teens and high-risk pregnant women) and framed within broader risk and healthcare disparity reduction objectives that did not prioritize addressing the dual pathology. However, both interventions were based on psychological therapeutic models with robust empirical support.

Both papers described efforts to maintain and ensure the intervention’s accuracy. However, they differed in the qualifications required for those providing treatment. While most of the “pregnancy advisors” in Joseph et al.’s study [51] held master’s degrees in health-related disciplines (e.g., psychology or nursing),“family health educators” in Barlow et al.’s study [39] only needed to meet the minimum requirement of a high school diploma. Likewise, the studies differed in the length of the intervention, ranging from8sessions [51] to 43 sessions [39].

In both cases, baseline measurements were taken before the intervention, and several follow-up measurements (between two and eight) were conducted afterward. Regarding the effect sizes and results, Barlow et al. [39] reported that after an educational intervention involving 43 structured lessons, there was a decrease in depressive symptoms (effect size = 0.16) and lower past-month use of marijuana (odds ratio= 0.65) and other illegal drugs (odds ratio = 0.67). In contrast, Joseph et al. [51] did not report specific measures of the reduction in depressive symptoms and smoking they assessed; instead, this information appeared together with other identified risk factors and was reported by the number of risk factors resolved after the intervention. Therefore, although the results after the intervention seem favorable, in this case, we cannot specifically determine whether depressive symptoms and tobacco consumption declined.

Regarding the rest of the full-text articles that do not meet the criteria of the systematic review (Table 1), it is worth noting that there are studies focusing on the development of therapeutic interventions to promote the well-being of women during the perinatal period. Specifically, four publications aim to reduce unintended pregnancies in at-risk populations (for example, adolescents) [48,49,69,82]. Additionally, 13 publications aim to improve the conditions of childbirth itself and to enhance the postpartum outcomes [34,36,40,42,43,65,66,70,71,73,77,80,87]. Furthermore, seven studies reported the effect of an intervention on variables other than the focus of this systematic review, such as the acceptance of the intervention, assessment of the implementation costs, or the intervention’s retention capacity [47,53,59,62,72,75,86]. Finally, it is worth noting that a great number of studies addressing substance use with RCTs also consider other aspects of the well-being of pregnant women, such as family cohabitation, employment integration, and accessibility to healthcare resources. However, they do not report measures of psychopathology; hence, it cannot be considered that they address dual pathology [38,47,52,57,58,67,68,76,79,85,88,90].

## 4. Discussion

The mental well-being of women throughout the perinatal phase is paramount to the overall health of society [91]. However, despite increased concern in recent years [10], there is still considerable ground to cover in the identification and management of perinatal mental health and substance use issues. Maternal mental health appears to be the new elephant in the room.

### 4.1. Regarding the Research Question

To determine the current status of psychological treatments for dual pathology in pregnant women, a systematic review was carried out following the PRISMA methodology. Despite some previous systematic reviews separately addressing treatment for mental health [3,28,29] and substance use problems [30,31,32] in pregnant women, evidence on treatments addressing both pathologies in an integrated manner is scarce.

Given the scarcity of research investigating the effectiveness of psychological interventions using rigorous methodologies such as randomized controlled trials (RCTs), particularly concerning both mental health and substance abuse in pregnant women, along with the diverse range of objectives and outcome measures utilized across studies, arriving at definitive conclusions regarding the efficacy of such interventions presents considerable complexity. The inherent variability in the types of interventions studied, the specific mental health or substance abuse issues addressed, and the outcome measures employed further complicates efforts to generalize the findings and establish clear guidelines for clinical practice.

### 4.2. Looking beyond the Final Results

Given the bleak panorama presented by this systematic review, we must consider the works excluded from the final selection (Table 1). Notably, we did find publications on various interventions addressing mental health and substance use problems in pregnancy independently. As revealed in the previous reviews mentioned above, other interventions ranged from Reiki, reflexology, and acupuncture to physical activity, social support, pharmacotherapy, and transcranial magnetic stimulation, in addition to psychoeducational and psychological interventions.

These findings are promising, although they should be viewed with caution as far as intervention in dual pathology is concerned. Implementing a methodological approach that isolates specific problems during intervention testing undoubtedly enhances the validity of research studies. However, this focused approach also introduces challenges when attempting to generalize findings to real-world clinical settings. By isolating individual problems, such as mental health or substance use issues, interventions may overlook the complex interplay and comorbidity often present in clinical populations, thus limiting their applicability to and effectiveness in addressing the multifaceted needs of patients. Moreover, this isolating approach perpetuates a reductionist view of individuals as collections of independent components rather than holistic beings. By treating mental health and substance use as separate entities, this approach reinforces the traditional siloed model of care, hindering efforts to integrate psychiatric and psychological services into broader healthcare systems [92]. This fragmentation not only undermines the delivery of comprehensive care but also perpetuates the stigma surrounding mental health and substance abuse, further marginalizing individuals seeking support and treatment [93]. Therefore, moving towards a more integrated and holistic approach to care is essential to effectively addressing the complex and interconnected needs of patients.

It is also noteworthy that the number of papers decreases when restricting the search to psychological interventions. It is striking that none of the reviews to date employed this as a selection criterion. While this gap may stem from a genuine interest in knowing the full range of tools for addressing mental health or substance problems in pregnant women, it raises concerns about the thoroughness of evaluating psychological intervention efficacy in this context. Rigorous evaluation of psychological interventions is crucial to informing evidence-based practice and ensuring the delivery of high-quality care.

Finally, it is striking that while the treatments described in the present review are psychological in nature, they are not necessarily provided by psychology/psychiatric professionals. Instead, social workers, educators, nurses, or primary care physicians are trained to deliver these interventions, which are typically brief. These proposals aim to improve accessibility by allowing the professionals commonly encountered by pregnant women to offer brief, focused, and inexpensive interventions, who can also perform screening functions. Given the limited access to treatment for pregnant women described in the literature [94,95,96], studying this type of approach is crucial.

However, an evident gap exists in the literature concerning rigorous evaluation of the psychological interventions delivered by mental health professionals, such as psychologists or psychiatrists, in cases necessitating specialized care for pregnant women. This critical gap in the research poses a significant challenge, as it hampers clinical decision-making processes and undermines the ability to deliver optimal care to this vulnerable population. This deficiency becomes even more concerning when considering that pregnant women, particularly those grappling with depression and anxiety during the perinatal period, express a clear preference for psychological interventions over alternative modalities such as primary care, nursing, or religious or social support [97]. By neglecting to rigorously evaluate the psychological interventions provided by trained mental health professionals, we risk failing to meet the preferences and needs of pregnant women seeking specialized care for mental health concerns during pregnancy.

Addressing this gap in the literature is imperative to enhance the quality and effectiveness of mental healthcare for pregnant women. By conducting rigorous evaluations of the psychological interventions delivered by mental health specialists, we can ensure that evidence-based practices are implemented, thus improving clinical outcomes and meeting the diverse needs of pregnant women experiencing mental health challenges during the perinatal period.

### 4.3. Limitations

This work has some limitations that require attention. First, we must be cautious about the conclusions drawn since some relevant studies may have been left out. Likewise, although the review process involved two independent reviewers and a third reviewer who checked the results, biases from the research group may have influenced the selection process. However, adherence to the PRISMA protocol for systematic reviews ensures some standardization of procedures used. Second, due to the objective of the review, the inclusion and exclusion criteria had to be very specific, culminating in the selection of only two papers and thereby revealing the scarcity of the available evidence. The narrowness of the selection criteria may impact the external validity of the current study. Many integrated interventions for substance use reduction also encompass other variables related to well-being and mental health. However, publications on these interventions may not have systematically reported measures of mental health, thus falling outside the scope of this review. For example, in the U.S.A., the Substance Abuse and Mental Health Services Administration (SAMHSA) has funded projects in this area for years through the Pregnant and Postpartum Women’s program (PPW) [98]. However, despite these programs being focused on evidence-based interventions, the published papers on these programs are not RCTs [90], thus falling outside the scope of the present systematic review. This limitation did not allow us to draw conclusions about the treatments evaluated to date, although it did permit us to illustrate the situation in which we currently find ourselves.

A final limitation worth noting is that throughout the entire study, exclusive reference is made to pregnant women, thereby excluding the spectrum of pregnant people who do not fully identify with the female gender. All the literature consulted, as well as the texts included in the systematic review, make reference solely to pregnant women; therefore, we cannot ascertain the degree of generalization of the results found in this study to that population group.

## 5. Conclusions and Future Directions

When addressing the complex needs of pregnant women grappling with dual pathology, it becomes evident that we are confronting a profound health challenge with far-reaching implications for both the expectant mother and the developing baby. Despite initial assumptions, the prevalence of dual pathology in pregnant women is alarmingly high [9], with serious consequences, including an increased risk of suicide and other adverse outcomes [99]. Recognizing the gravity of this issue, there is an urgent need to develop specialized and integrated interventions tailored to addressing the unique needs of this vulnerable population [7].

As highlighted in the present review, evidence-based interventions are paramount to effectively addressing the multifaceted challenges associated with dual pathology during pregnancy. Drawing from established cognitive-behavioral models of intervention and the transtheoretical model of behavior change, these interventions should be grounded in rigorous scientific evidence to ensure their effectiveness and appropriateness for pregnant women [39,51]. Moreover, evaluation of these interventions using randomized controlled trials (RCTs) is imperative to establish their level of efficacy and inform evidence-based practice. By subjecting interventions to rigorous evaluation methodologies, we can ascertain their effectiveness in improving maternal and fetal outcomes, thus guiding clinical decision-making and ensuring the delivery of high-quality care.

Furthermore, it is essential for these interventions to be delivered also by trained mental health specialists, both in clinical practice and during RCTs. This ensures that the interventions are implemented with the necessary expertise and rigor, facilitating the generalization of the results to specialized consultations and enhancing the scalability and sustainability of effective interventions.

In conclusion, the development and implementation of specialized and evidence-based interventions represent crucial steps towards addressing the serious health challenges faced by pregnant women with dual pathology. By integrating scientific evidence, rigorous evaluation methodologies, and specialized expertise, we can enhance the quality of care provided to this vulnerable population, ultimately improving maternal and fetal health outcomes and fostering healthier pregnancies.

## Figures and Tables

**Figure 1 ijerph-21-00392-f001:**
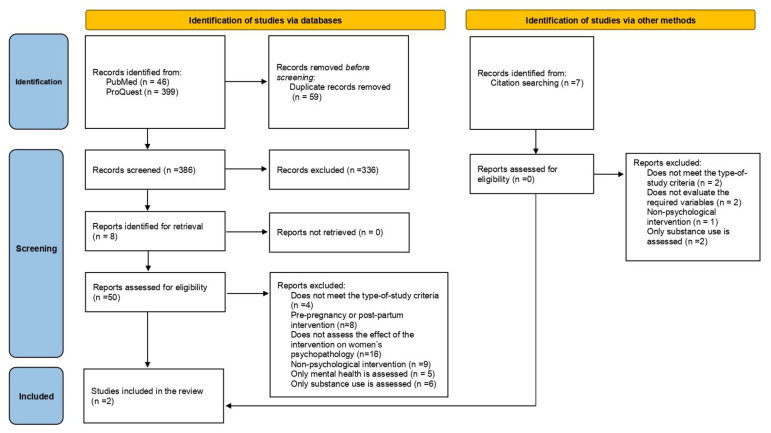
Flow diagram for the systematic review according to the PRISMA 2020 statement (Page, 2021). Adapted from [33].

**Table 1 ijerph-21-00392-t001:** Summary of full-text publications reviewed and adjustment to systematic review criteria.

Publication/Authorship and Year	Meets the Criteria of the Type of Study	Participants Are Pregnant Women	Measures *	The Described Intervention Is Psychological	Addresses Dual Pathology
Asadzadeh et al., 2020 [34]	X				
Surkan et al., 2020 [35]	X	X			
Baas et al., 2017 [36]	X	X			
Baas et al., 2022 [37]	X	X			
Bar-Zeev et al., 2017 [38]	X	X			
Barlow et al., 2015 [39]	X	X	X	X	X
Barros et al., 2022 [40]	X	X			
Bayat et al., 2021 [41]	X	X	X	X	
Burduli et al., 2021 [42]	X	X			
Coleman-Cowger, 2012 [43]					
Crockett et al., 2008 [44]	X	X			
Fischer, 2000 [45]					
Foroughinia et al., 2020 [46]	X	X	X		
Gray et al., 2017 [47]	X	X			
Hanson and Jensen, 2015 [48]	X				
Ingersoll et al., 2003 [49]	X				
Janssen et al., 2012 [50]	X	X	X		
Joseph et al., 2009 [51]	X	X	X	X	X
Jussila et al., 2020 [52]	X	X	X	X	
Kennedy et al., 2004 [53]	X	X			
Kenyon et al., 2016 [54]	X	X	X	X	
Kingston et al., 2015 [55]	X	X			
Kurzeck et al., 2021 [56]	X	X	X		
Lander et al., 2015 [57]	X	X	X	X	
Larden et al., 2005 [58]	X	X	X		
Lee King et al., 2015 [59]	X	X			
Maleki et al., 2021 [60]					
Martín Maldonado-Duran et al., 2000 [61]					
Miles et al., 2001 [62]					
Moradi et al., 2022 [63]	X	X	X		
Nasiri et al., 2019 [64]	X	X	X		
Navas et al., 2021 [65]	X	X	X		
Olds et al., 2019 [66]	X	X	X	X	
Parrish et al., 2012 [67]	X				
Peles et al., 2014 [68]	X	X	X	X	
Penberthy et al., 2013 [69]	X				
Popo et al., 2017 [70]	X	X	X		
Racine et al., 2021 [71]	X				
Rezaie et al., 2021 [72]	X	X			
Rimehaug et al., 2019 [73]	X	X			
Stone et al., 2017 [74]	X	X	X		
Strantz and Welch, 1995 [75]	X	X			
TzilosWernette et al., 2011 [76]	X	X	X	X	
Veringa et al., 2016 [77]	X	X			
Vieten and Astin, 2008 [78]	X	X	X	X	
von Sternberg et al., 2018 [79]	X				
Weinreb et al., 2018 [80]	X	X	X	X	
Wilder and Winhusen, 2015 [81]	X	X	X		
Wilton et al., 2013 [82]	X				
Windsor et al., 2017 [83]	X	X	X	X	
Windsor et al., 2011 [84]	X	X	X	X	
Windsor et al., 2014 [85]	X	X			
Xu et al., 2017 [86]	X	X			
YalnızDilcen and Genc, 2019 [87]	X	X			
Yonkers et al., 2009 [88]	X	X	X	X	
Zemestani and Fazeli Nikoo, 2020 [89]	X	X	X	X	
Bray et al., 2022 [90]					

Note. * shows measures of the effect of an intervention on the psychopathology of pregnant women.

**Table 2 ijerph-21-00392-t002:** Main characteristics of the studies finally included in the systematic review.

Study	Sample Characteristics (Country, Type of Sample, N, and Age)	Objective	Method (Type of Study, Evaluations, Variables, Instruments)	Description of the Intervention	Main Results	Main Conclusions
Barlow et al. (2015) [39]	U.S.A. Eligible participants were expectant American Indian women from four southwestern reservation communities, recruited from Indian Health Service clinics; Women, Infants, and Children nutrition programs; and schools and by word of mouth. Total N = 322; age: M = 18.1 (SD = 1.5) Intervention group N = 159; age: M = 18.2 (SD = 1.4) Control group N = 163; age: M = 18,1 (SD = 1.6) >18 years N = 186 (57.8%)	To evaluate the intervention’s effects on parental competence (parenting knowledge, locus of control, stress, and behaviors) and maternal behavioral problems that impede effective parenting through early childhood (0 to 36 months postpartum). The secondary aims were to evaluate the intervention’s effects on early childhood emotional and behavioral outcomes.	Multisite, randomized (1:1), parallel-group trial. Participants were randomly assigned to the Family Spirit intervention plus optimized standard care or optimized standard care alone. Maternal and child outcomes were evaluated at 28 and 36 weeks gestation and 2, 6, 12, 18, 24, 30, and 36 months postpartum. Variables and instruments: Parental competence: a 30-item maternal self-report created ad hoc Home environment: Home Observational Measure of the Environment (HOME) Maternal emotional and behavioral functioning: The Center for Epidemiological Studies Depression scale (CES-D); the Achenbach System of Empirically Based Assessments Youth self-report Quantity and frequency of alcohol and drug use: the drug use subscale of Voices for Indian Teens Children’s emotional and behavioral outcomes: the Infant-Toddler Social and Emotional Assessment	The Family Spirit intervention: a total of 43 structured lessons in a culturally congruent format, focused on reducing behaviors associated with early childhood behavior problems and addressing maternal behavior and mental health problems (including substance use and externalizing and internalizing behaviors). Lessons were delivered one on one in the participants’ homes, weekly through the third trimester of pregnancy, biweekly until 4 months postpartum, monthly between 4 and 12 months postpartum, and bimonthly between 12 and 36 months postpartum. Interventionists were trained people required to have a minimum of a high school diploma, 2 years of job-related education or work experience, and the ability to speak a Native language and English.	From pregnancy to 36 months postpartum, the participants in the intervention group showed significantly greater parenting knowledge (effect size = 0.42) and parental loci of control (effect size = 0.17), fewer depressive symptoms (effect size = 0.16) and externalizing problems (effect size = 0.14), and lower past-month use of marijuana (odds ratio = 0.65) and illegal drugs (odds ratio = 0.67).	The home-delivered intervention promotes a more effective parenting style, improves depressive symptoms, and reduces the mother’s substance use in a population with fewest resources
Joseph et al. (2009) [51]	U.S.A. Eligible participants were expectant African American or Latina women recruited from six prenatal care sites in Washington, DC. Total sample N: 953; age: M = 24.6 (SD = 0.2) Intervention group N = 470; Age: M = 24.4 (SD = 0.3) Usual care group N = 483; age: M = 24.8 (SD = 0.3)	To evaluate the efficacy of a primary care intervention targeting pregnant African American women and focusing on psychosocial and behavioral risk factors for poor reproductive outcomes (cigarette smoking, second-hand smoke exposure, depression, and intimate partner violence).	Multisite, randomized, parallel-group trial. Participants were randomly assigned to an intervention or usual care group. Multiple imputation methodology was used to estimate missing data. The maternal outcomes were evaluated at baseline (<28 weeks of gestation), first follow-up (second trimester), and second follow-up (third trimester). Variables and instruments: cigarette smoking and secondhand smoke exposure: questionnaire created ad hoc Depression: the Hopkins Symptom Check List Intimate partner violence: the Revised Conflict Tactics Scale	A behavioral intervention: Eight individually tailored counseling prenatal sessions based on the Smoking Cessation or Reduction in Pregnancy Trial (SCRIPT), the transtheoretical model of behavior change, the “pathways to change” self-help manual and cognitive-behavioral therapy (clinic-based) were adapted from evidence-based interventions. The intervention sessions were delivered immediately before or after routine prenatal care in the prenatal care clinics of the participants. Interventionists were trained people required to have a degree in disciplines such as psychology and experience in interpersonal counseling, health education, or behavioral change.	Two approaches to quantify behavioral changes: contrasting the distribution of the number of risks reported by participants in the two groups and quantifying within-person change over time. The number of risks did not differ between the intervention and usual care groups at baseline, the second trimester, or the third trimester. The women in the intervention group more frequently resolved some or all of their risks than did the women in the usual care group (odds ratio = 1.61; [95% CI 1.08–2.39]; *p* = 0.021).	An 8-session cognitive-behavioral intervention, compared to usual treatment, can help women at psychosocial risk to reduce it.

## Data Availability

The collected data are available on Sysrev.com (accessed on 20 February 2024.).

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
