# Peer review of "The Elephant in the Room: A Systematic Review of the Application and Effects of Psychological Treatments for Pregnant Women with Dual Pathology (Mental Health and Substance-Related Disorders)"

_ijerph, 2024, doi:10.3390/ijerph21040392_

Round 1

Reviewer 1 Report

Comments and Suggestions for Authors

The authors identify an important area of concern, which is the limited evidence base regarding interventions for co-occurring perinatal SUD and mental health conditions.  However, it seems that the review criteria were so narrow as to have limited external validity.  Many integrated SUD treatment programs for perinatal populations do include extensive mental health treatment supports because co-occurrence of these disorders is nearly universal.  However, publications about these programs may not have systematically measured mental health outcomes.  It seems that the authors over-generalize in the beginning of the discussion section of the paper, saying "Despite increased concern in recent years, the identification and management of perinatal mental health conditions have not received sufficient emphasis."  This statement is misleading-- while the authors found a paucity of publications addressing interventions for co-occurring disorders, there is a large evidence base regarding psychological interventions for mental health disorders themselves.

Comments on the Quality of English Language

English in the abstract needs some editing-- use of language is atypical for a scientific publication and almost seems as if it was translated electronically.  For example, the phrase "is a big concern" is unusual for a formal manuscript.  Would also recommend that the authors consider using gender inclusive language as not all people experiencing pregnancy are female or identify as female.  

Author Response

Dear Reviewer #1

Thank you for the attention with which you have examined our manuscript and for your valuable comments, which have led to the modifications that, following your instructions, are set out below:

1- The authors identify an important area of concern, which is the limited evidence base regarding interventions for co-occurring perinatal SUD and mental health conditions.  However, it seems that the review criteria were so narrow as to have limited external validity.  Many integrated SUD treatment programs for perinatal populations do include extensive mental health treatment supports because co-occurrence of these disorders is nearly universal.  However, publications about these programs may not have systematically measured mental health outcomes. 

Answer to the reviewer: Thank you very much for pointing out this issue. This has been added as a limitation of the work:

The narrowness of the selection criteria may impact the external validity of the current study. Many integrated interventions for substance use reduction also encompass other variables related to well-being and mental health. However, publications on these inter-ventions may not have systematically reported measures of mental health, thus falling outside the scope of this review. For example, in the U.S.A. the Substance Abuse and Mental Health Services Administration (SAMHSA) has funded projects in this area for years through the Pregnant and Postpartum Women’s program (PPW) [98]. However, despite these programs being focused on evidence-based interventions, the published papers on these programs are not RCTs [90], thus falling outside the scope of the present systematic review.

2- It seems that the authors over-generalize in the beginning of the discussion section of the paper, saying "Despite increased concern in recent years, the identification and management of perinatal mental health conditions have not received sufficient emphasis."  This statement is misleading-- while the authors found a paucity of publications addressing interventions for co-occurring disorders, there is a large evidence base regarding psychological interventions for mental health disorders themselves.

Answer to the reviewer: Thanks for bringing this to our attention, the sentence has been modified to:

However, despite increased concern in recent years [10], there is still considerable ground to cover in the identification and management of perinatal mental health and substance use issues.

3- Comments on the Quality of English Language: English in the abstract needs some editing-- use of language is atypical for a scientific publication and almost seems as if it was translated electronically.  For example, the phrase "is a big concern" is unusual for a formal manuscript. 

Answer to the reviewer: Thank you for highlighting this. The abstract has been professionally reviewed.

4- Would also recommend that the authors consider using gender inclusive language as not all people experiencing pregnancy are female or identify as female.  

Answer to the reviewer: We are grateful to the reviewer for bringing this matter to our attention. Indeed, there is a segment of the pregnant population that does not identify with the female gender. However, all the references used in the manuscript exclusively address pregnant women, and it is impossible for us to ascertain whether this is due to a lack of assessment of gender orientation or if indeed all the women in the sample identified themselves as women. For this reason, we have chosen to maintain the original text. Nonetheless, recognizing this limitation, we have included the following paragraph in the limitations section:

A final limitation worth noting is that throughout the entire study, exclusive reference is made to pregnant women, thereby excluding the spectrum of pregnant people who do not fully identify with the female gender. All the literature consulted, as well as the texts included in the systematic review, make reference solely to pregnant women; therefore, we cannot ascertain the degree of generalization of the results found in this study to that population group.

Reviewer 2 Report

Comments and Suggestions for Authors

Review of ijerph-2897765

This is an interesting paper that reviews psychological treatments for pregnant women with both mental health and substance use problems.  The paper is a systematic review of only studies that had a RCT design that used psychological interventions.  The paper addresses a very important and under-studied area.  The introduction outlines the issues in this area. The methods were clear and easy to follow.  The fact that they only reviewed 2 studies limits the conclusions and these are acknowledged by the authors.

While the authors followed standardized procedures for identifying studies, they did not capture several published studies or entire program areas that focus on pregnant and postpartum women with mental health and substance use problems.  For example, in the U.S. the Substance Abuse and Mental Health Services Administration (SAMHSA) has funded projects in this area for years through the Pregnant and Postpartum Women’s program (PPW).  There have been several papers published on these programs. They are not RCTs, but the programs focus on evidence-based psychological, psychiatric and case management interventions.  To receive funding from the PPW program, only evidence-based interventions that have been developed for this population are allowed. 

Overall, this paper makes a good contribution, but it would be much more interesting and complete if they did a review of the other studies listed in Table 1 and expanded that table to include SAMHSA PPW research. There are data in those studies that examine outcomes for the PPW participants. 

References:

Bray, J. H., Scamp, N., Zaring-Hinkle, B., Tucker, K., & Cain, M. (2022). MIRRORS Program: Helping pregnant and postpartum women and families with substance use problems. Substance Abuse, 43(1), 792-800. DOI: 10.1080/08897077.2021.2010254

Clark HW. Residential substance abuse treatment for pregnant and postpartum women and their children: treatment and policy implications. Child Welfare. 2001;80(2).

Author Response

Dear Reviewer #2

Thank you very much for your analysis of our work. We have meticulously taken into account your suggestions for improvement as set out below:

1- This is an interesting paper that reviews psychological treatments for pregnant women with both mental health and substance use problems.  The paper is a systematic review of only studies that had a RCT design that used psychological interventions.  The paper addresses a very important and under-studied area.  The introduction outlines the issues in this area. The methods were clear and easy to follow.  The fact that they only reviewed 2 studies limits the conclusions and these are acknowledged by the authors.

Answer to the reviewer: Thank you for your feedback

2- While the authors followed standardized procedures for identifying studies, they did not capture several published studies or entire program areas that focus on pregnant and postpartum women with mental health and substance use problems.  For example, in the U.S. the Substance Abuse and Mental Health Services Administration (SAMHSA) has funded projects in this area for years through the Pregnant and Postpartum Women’s program (PPW).  There have been several papers published on these programs. They are not RCTs, but the programs focus on evidence-based psychological, psychiatric and case management interventions.  To receive funding from the PPW program, only evidence-based interventions that have been developed for this population are allowed. 

Answer to the reviewer:  We agree with the reviewer. Thank you for bringing this issue to our attention. The limitations of the paper have been deepened and the following paragraph added:

. The narrowness of the selection criteria may impact the external validity of the current study. Many integrated interventions for substance use reduction also encompass other variables related to well-being and mental health. However, publications on these inter-ventions may not have systematically reported measures of mental health, thus falling outside the scope of this review. For example, in the U.S.A. the Substance Abuse and Mental Health Services Administration (SAMHSA) has funded projects in this area for years through the Pregnant and Postpartum Women’s program (PPW) [98]. However, despite these programs being focused on evidence-based interventions, the published papers on these programs are not RCTs [90], thus falling outside the scope of the present systematic review.

3- Overall, this paper makes a good contribution, but it would be much more interesting and complete if they did a review of the other studies listed in Table 1 and expanded that table to include SAMHSA PPW research. There are data in those studies that examine outcomes for the PPW participants. 

References:

Bray, J. H., Scamp, N., Zaring-Hinkle, B., Tucker, K., & Cain, M. (2022). MIRRORS Program: Helping pregnant and postpartum women and families with substance use problems. Substance Abuse, 43(1), 792-800. DOI: 10.1080/08897077.2021.2010254

Clark HW. Residential substance abuse treatment for pregnant and postpartum women and their children: treatment and policy implications. Child Welfare. 2001;80(2).

Answer to the reviewer: Thank you again for highlighting this. Two paragraphs have been included in the results section to provide further analysis of the studies listed in Table 1:

Regarding the rest of the full-text articles that do not meet the criteria of the systematic review (Table 1), it is worth noting that there are studies focusing on the development of therapeutic interventions to promote the well-being of women during the perinatal period. Specifically, four publications aim to reduce unintended pregnancies in at-risk populations (for example, adolescents) [48,49,69,82]. Additionally, 13 publications aim to improve the conditions of childbirth itself and to enhance the postpartum out-comes  [34,36,40,42,43,65,66,70,71,73,77,80,87]. Furthermore, 7 studies reported the effect of an intervention on variables other than the focus of this systematic review, such as the acceptance of the intervention, the assessment of implementation costs, or the interven-tion's retention capacity [47,53,59,62,72,75,86]. Finally, it is worth noting that a great amount of studies addressing substance use with RCTs also consider other aspects of the well-being of pregnant women, such as family cohabitation, employment integration, and accessibility to healthcare resources. However, they do not report measures of psy-chopathology, hence, it cannot be considered that they address dual pathology [38,47,52,57,58,67,68,76,79,85,88,90].

One of the references suggested by the reviewer has been added to Table 1, which has also meant updating the procedure:

Finally, prior to the publication of this manuscript, in February, 2024, the search was conducted again looking for newly published papers and, after reviewing the title and abstract of 110 new references, no new results were found. Cross-reference check until March, 2024, results in one extra study included for full-text review (see Table 1).

Additionally, the two suggested references have been used to expand on the limitations and strengthen the reviewer's argument, as previously noted.